# The Effect of Quenching and Partitioning (Q&P) Heat Treatment on the Microstructure and Mechanical Properties of High Boron Steel

**DOI:** 10.3390/ma14061556

**Published:** 2021-03-22

**Authors:** Zhao Li, Run Wu, Mingwei Li, Song-Sheng Zeng, Yu Wang, Tian Xie, Teng Wu

**Affiliations:** 1Key Laboratory for Ferrous Metallurgy and Resources Utilization of Ministry of Education, Wuhan University of Science and Technology, Wuhan 430081, China; lizhaodoc@hotmail.com (Z.L.); li1725217198@163.com (M.L.); xt_zoe@163.com (T.X.); wuteng@wust.edu.cn (T.W.); 2Valin ArcelorMittal Automotive Steel Co., Ltd., Loudi 417009, China; zsscsu@sina.com; 3Mark Wainwright Analytical Centre, University of New South Wales, Sydney, NSW 2052, Australia; yu.wang@unsw.edu.au

**Keywords:** high boron steel, Q&amp, P heat treatment, retained austenite, TRIP effect, wear resistance

## Abstract

High boron steel is prone to brittle failure due to the boride distributed in it with net-like or fishbone morphology, which limit its applications. The Quenching and Partitioning (Q&P) heat treatment is a promising process to produce martensitic steel with excellent mechanical properties, especially high toughness by increasing the volume fraction of retained austensite (RA) in the martensitic matrix. In this work, the Q&P heat treatment is used to improve the inherent defect of insufficient toughness of high boron steel, and the effect mechanism of this process on microstructure transformation and the change of mechanical properties of the steel has also been investigated. The high boron steel as-casted is composed of martensite, retained austensite (RA) and eutectic borides. A proper quenching and partitioning heat treatment leads to a significant change of the microstructure and mechanical properties of the steel. The net-like and fishbone-like boride is partially broken and spheroidized. The volume fraction of RA increases from 10% in the as-cast condition to 19%, and its morphology also changes from blocky to film-like. Although the macro-hardness has slightly reduced, the toughness is significantly increased up to 7.5 J·cm^−2^, and the wear resistance is also improved.

## 1. Introduction

In recent years, high boron steel gains many scholars’ attention and has been widely used in the production of related workpieces in the fields of automotive industry, mining machinery, machine tools, etc., for its low cost [1,2], good hot workability [3,4,5], and high hardness. Boride (mainly Fe_2_B), as a strengthening phase in high boron steel, is superior to carbide in hardness (1450–1800 HV) and modulus of elasticity, which ensures that the steel has higher hardness and better wear resistance than white cast iron and high chromium cast iron [6,7]. However, due to the unique characteristics of boron in the microstructure, i.e., nonequilibrium boron segregation to austenite grain boundaries before quenching [8], the boride is usually distributed in net-like or fish-bone morphology in the solidified microstructure, which will seriously damage the continuation of the matrix, resulting in the inherent defect of insufficient toughness in high boron steel. Furthermore, Fe_2_B exhibits serious brittleness because of the weak B–B bond along [2] crystal orientation [9,10,11,12,13], which will make it easily peeled off during the wear process and deteriorate the wear resistance of the steel [14]. These extremely limit the application of high boron steel on the impacting condition.

Alloying and high-temperature heat treatment are effective ways to improve the toughness of high boron steel. The Fe_2_B grains can be refined by Ti and Ni addition in the alloy and decomposition of net-like Fe_2_B is facilitated by Ce addition [15,16]. The distribution of boride is improved by modification of the alloy molten with V, Ti, and RE-Mg [17], which improves the toughness of a high boron iron-based alloy. Moreover, the addition of a small amount of Nb or Mo can help to enhance boron’s hardenability in the steel, and prevent boron from forming boron carbides in the grain boundary [18,19,20,21], thereby reducing the obstructing effect of solute boron on ferrite nucleation [22]. Except for these, the net-like Fe_2_B can be disconnected by high temperature austenization [23,24]. Such approaches mentioned above like alloying and high temperature heat treatment are mainly focused on ameliorating the morphology of the boride, but there are few reports about improving matrix toughness. Attractive combination of strength and ductility of steel can be achieved by inserting a certain fraction of retained austensite (RA) in stronger martensite phase through Q&P heat treatment [25,26,27]. Therefore, Q&P heat treatment is a very novel and promising method for improving the microstructure and performance of high boron steel.

In the present study, the as-cast and Q&P heat-treated microstructures of high boron steel have been observed. In addition, the effect of Q&P heat treatment on the morphology and distribution of RA in the steel has been analyzed. Simultaneously, the macro-hardness, impact toughness, and wear resistance of high boron steel in the as-cast condition and Q&P heat-treated have been investigated. Subsequently, the wear morphologies have been observed and the wear mechanism has been discussed for the steel.

## 2. Materials and Methods

The high boron steel was fabricated by induction melting in a 50kg vacuum furnace with its chemical composition shown in Table 1. The weight fraction of boron in the steel is 1.6 wt.% to ensure the formation of the wear-resistant phase Fe_2_B. In order to stimulate the role of carbon and manganese in improving the toughness of the matrix during the partitioning process, the content of these two elements were set to 0.45 wt.% and 0.6 wt.%, respectively. The modifier addition of 0.5 wt.% RE-Mg was carried out in an Aluminum foil package prior to casting into metallic molds.

Q&P heat treatment process was realized using the combination of electric furnace and the salt baths, illustrated in Figure 1. First, the samples of the high boron steel were austenitized at 1050 °C for 2 h in the electric furnace so as to improve the morphology of eutectic borides. Subsequently, the samples were moved to the first salt bath, which was set up at the quench temperature (Tq). The quenching time (tq) was set to 30 s, 60 s, 90 s, and 120 s, respectively. When the quenching temperature was reached, samples were immersed in another salt bath to realize the partitioning step (400 °C for 60 s). After partitioning the samples, water was quenched to room temperature and then final samples were obtained.

Tq was set up as following: the temperature was calculated by Koistinen-Marburger equation [28]:(1)$$f_{M}=1-\exp[-0.011(\mathrm{Ms}-\mathrm{Tq})]$$

Since the high boron steel contains a large volume fraction of borides, the K-M equation was appropriately changed as in Equation (2):(2)$$f_{M}=\left(1-V_{B}\right)\{1-\exp[-0.011(\mathrm{Ms}-\mathrm{Tq})]\}$$
where *f_M_* is the percentage of quenched martensite, and *V_B_* is the volume fraction of boride in the structure, which is 21.3% measured by the metallo-graphical method in the as-cast microstructure in the steel. Ms point is 225 °C measured by Gleeble-3500 thermal simulation tester (DATA SCIENCES INTERNATIONAL, INC., St. Paul, MN, USA). The calculation result of Tq is shown in Figure 2. Therefore, Tq is chosen at about 160 °C.

Samples for microstructural characterization were grinded, polished, and etched in a 4% nital solution. Microstructures were observed by ZEISS metallographic microscope (Carl Zeiss AG, Oberkochen, Germany) and FEI Nova 400 Nano SEM field emission scanning electron microscope (Royal Dutch Philips Electronics Ltd., Amsterdam, Netherlands) after etching in a 4% nital solution. Transmission electron microscope (TEM), JEM-2100F (JEOL, Tokyo, Japan), was also used to characterize the microstructure of the samples, which were prepared by grinding and polishing down to a thickness of about 50 μm, followed by double-jet thinning at −25 °C with operating voltage of 32 V using 10 vol.% perchloric acid solution.

The phase composition was measured by X-ray diffraction (XRD) on a miniFlex 600 X-ray diffractometer (Lijing Scientific Instrument Co., Ltd., Shanghai, China) with CuKα radiation with 40 kV and 30 mA. Samples were scanned in a 2θ range from 40° to 101° using a step size of 0.01°. The volume fraction of retained austenite (RA) was also measured by a LakeShore 480 magnetic measuring instrument (Linkphysics Corporation, Shanghai, China). Electron backscattering diffraction (EBSD) analysis was conducted based on the same SEM field of the samples after being vibration-polished. The data were acquired at an accelerating voltage of 20 kV, a working distance of 15 mm, a tilt angle of 70°, and a step size of 40 nm.

The macro-hardness of the samples was measured by an HRS-150 Rockwell hardness tester (Huayin Testing Instrument Co., Ltd., Laizhou, China) and the final value is the average of ten readings. Impact tests were performed by a JB-50 impact tester (Jianyi Experiment Equipment Co., Ltd., Wuxi, China) at room temperature. The samples had no notches with a size of 10 × 10 × 55 mm^3^ and the values reported were the averages of three tests. Dry sliding wear tests were done by the MM-2000 wear tester (Zhengli Balancing Machine Co., Ltd., Kalgan, China). The samples for the wear test with a size of 6 × 7 × 15 mm^3^, which were cut from steel after Q&P treatment and in the as-cast condition. The GCr15 with hardness of 60–62 HRC was selected as the lower specimen and the detected samples were set as upper ones. The test parameters are an applied load of 350 N, a total time of wear being 3 h, and the speed of the lower specimen being 200 r/min. NM500 (51.3 HRC) was selected as the reference specimen. Each kind of sample and reference specimen were all subjected to three wear tests, and the mass loss were measured by the electronic balance with an accuracy of 0.1 mg. The final value is the average of three measurements, and the deviation between the maximum value and the minimum value was counted.

## 3. Results and Discussion

### 3.1. Microstructural Characterization

#### 3.1.1. Microstructure of High Boron Steel in the As-Cast Condition

Figure 3 exhibits the optical and scanning electron micrographs of the high boron steel in the as-cast condition. From Figure 3a,b, it can be seen that the high boron steel is composed of matrix and a eutectic microstructure. According to the XRD spectrum (Figure 4), it is evident that the phases of the high boron steel are α-Fe, γ-Fe, M_2_B, and M_23_(B,C)_6_, where M represents Fe or Mn and γ-Fe phase indicates the existence of retained austenite (RA) in the microstructure. Martensite is identified as α-Fe. Furthermore, the borides in eutectic show net-like and rod-like morphology.

The microstructure resulted from the solidification of the high boron steel. The boron concentration of the remaining melt increases with the formation of dendrites due to the low solubility of boron in austenite, resulting in the formation of various eutectic borides around the dendrites. First, γ proeutectic transformation occurs as the liquid is cooled below the liquidus temperature. The γ grows in a form of dendrite and exclude boron atoms in liquid. When B-enriched molten steel reaches a eutectic point, eutectic reaction occurs: L→ γ + Fe_2_B. In addition, a very small amount of secondary boron carbide, M_23_(B,C)_6_, precipitates from the matrix during the cooling process. The formation of boride requires a large number of alloying elements. There is a strong p-d interaction between B and Fe, Mn atoms as two adjacent elements, which will form strong covalent bonds of Fe-B and Mn-B. Therefore, Mn will displace part of Fe and increase Mn content in borides around dendrites. After solidification, martensite transformation will occur in the austenite, and, finally, the matrix is composited of martensite laths and RA shown in Figure 3b.

#### 3.1.2. Microstructure of High Boron Steel after Q&P Heat Treatment

After being treated by the Q&P process, the phase composition of the steel is the same as that of as-cast condition, but the intensity of the austenite peak is significantly increased in the XRD diffraction pattern of the samples from tq 30 s to 120 s, as shown in Figure 4. However, the microstructures are clearly different. The microstructures of the high boron steel after Q&P heat treatment are shown in Figure 5, where Figure 5a,c,e,g are optical microstructures and Figure 5b,d,f,h are SEM microstructures.

The boride in eutectic is partially spheroidized and fractured in all the samples after austenitizing (1050 °C + 2 h). During solidification, the necking and weak joints occur in the boride branches by the influence of the modifier. By analysis of thermodynamics, the interface energy between necking and weak joints of the boride and matrix is larger than that between spherical boride and the matrix [17,29]. In the austenization process, the difference of the interface energy result is the morphology of the boride, which is partially changed to fracture-like and spheroid-like from net-like. The matrix microstructure in the dendrite is observed to be un-tempered martensite (UM) and tempered martensite (TM). In the sample’s quenched time (tq) of 30 s, a gray white structure is observed, which show lath martensite as seen in Figure 5a,b, called UM. In the one with tq for 60 s, a black structure with clear lath is observed in the matrix, which is called TM (Figure 5c,d).

Figure 6 exhibits the quantitative results of samples’ microstructure after Q&P heat treatment. The boride decreased slightly from 21.3% of the as-cast condition to approximately 18%, and remain nearly the same amount in the partitioning with the extension of tq due to M_2_B that will not change in the subsequent partitioning process, and the change of the matrix in the eutectic is consistent with that of the matrix structure of austenite dendrite. However, the volume fraction of TM gradually increases, which are 4% (tq = 30 s), 36% (tq = 60 s), and 57% (tq = 90 s) to 67% (tq = 120 s), respectively. Meanwhile, the volume fraction of UM gradually decreases with the volume fraction of 79% (tq = 30 s), 46% (tq = 60 s), 27% (tq = 90 s) to 15% (tq = 120 s).

The microstructure quenched from austenizing is detected to have transformation during the partitioning process. The carbon atoms are diffused from the martensite into RA, which improves its thermostability. Meanwhile, the carbides precipitates within the martensite, and a tempered martensite is finally obtained with the name of TM. The UM is the product of the martensitic transformation of RA. In the partitioning process, some of RA cannot be obtained as sufficient carbon to enhance its thermostability. This part of RA is transformed into martensite when the samples are cooled down to room temperature. The new formed martensite, called UM, show a gray-white microstructure.

The volume fraction of RA in the sample changes after Q&P heat treatment. It can be observed from Figure 7 that the fraction of RA increases from 4.2% to 19.7% as the tq increases. It is noted that the fraction of RA decreases from 10% in the as-cast condition to 4.3% when tq is 30 s. This is because the short tq results in a small volume fraction of martensite formed in the microstructure after quenching from austenizing, so that only a very small amount of carbon can be diffused into the RA during the subsequent partitioning at a temperature of 400 °C for 60 s. Therefore, only a small fraction of RA can be stabilized to room temperature. However, with the extension of tq, the martensite fraction in the microstructure increased after quenching, so that more carbon can be diffused into the RA during the partitioning process, which improved the stability of RA.

### 3.2. Distribution and Morphology of Retained Austenite in the High Boron Steel before and after Q&P Heat Treatment

The distribution of RA is characterized by EBSD, as shown in Figure 8. In the EBSD phase maps, Fe_2_B phase is blue, the RA phase is red, and the martensite phase is gray. The amount of RA is increased between the eutectic borides and in the dendrite by quenching from 60 s to 120 s. It is attributable to the increase of martensite content in the samples after quenching from austenizing, and, therefore, more carbon atoms can be diffused from the martensite to the adjacent austenite crystals in partitioning. The thermostability of RA is, therefore, enhanced by the increase of carbon content, which results in more RA stabilized at room temperature. Additionally, the more RA is observed in the matrix (Figure 8a,c) near to borides and especially in the eutectic microstructure (Figure 8b,d), making more uniform distribution of RA.

Difference of RA morphology is detected in the as-cast sample and in the Q&P treated ones by TEM, as shown in Figure 9. The blocky RA is observed in the eutectic microstructure of as-cast samples from Figure 9a,b, with a width of about 0.7 μm. However, after Q&P heat treatment, film-like RA in the width of about 30 nm appears between martensite laths in the samples with tq of 90 s and 120 s. The reasons are as following. The segregation of Mn and C in the as-cast microstructure during solidification, causing RA to exist in block morphology at the edge of the dendrite. The composition of the samples is homogenized by high-temperature austenitization in Q&P heat treatment, and the martensite can uniformly nucleate in the matrix, resulting in the formation of film-like RA between martensite laths. RA presenting film-like has better transformation stability compared with a blocky type, which tends to transform to martensite under a small stress and contributes little to the TRIP effect [30,31,32,33] and can hardly improve the mechanical properties of the steel.

### 3.3. Effect of Q&P Heat Treatment on Mechanical Properties of High Boron Steel

#### 3.3.1. Hardness and Impact Toughness of the High Boron Steel Treated by Q&P Heat Treatment

Table 2 lists hardness and impact toughness of samples treated at different tq. The hardness of the samples gradually decrease from 61.8 to 55.3 HRC with tq increases from 30 s to 120 s. Only the hardness of the sample treated at tq of 30 s is 61.8 HRC, which is higher than that of the as-cast condition (59.0 HRC). The hardness increase is due to the homogenization of composition and elimination of blocky RA after Q&P treatment. However, the decrease in hardness is the result of the increase of the soft phase RA in the microstructure with the extension of the tq.

The impact toughness of high boron steel is improved by Q&P heat treatment. Compared with the as-cast sample (3.7 J·cm^−2^), the toughness of samples treated at different tq all gets improved and reaches the maximum (7.5 J·cm^−2^) in the sample with tq of 90 s. Figure 10 shows the impact fracture surface of samples. It can be seen from Figure 10a that the dendrite zone is quasi-cleavage fracture, while there are clear boride pits in the eutectic zone, which are mostly left by intergranular fractures along the interface between the boride and the matrix. When tq is 90 s, as shown in Figure 10b, the dendritic zone is filled with dimples and fishbone boride is covered by the ductile matrix in the eutectic zone. The improvement of toughness is attributed both to the fracture or spheroidization of the boride and also to the increase of matrix toughness [34,35]. For high boron steel, cracks will occur and propagate at the interface between boride and matrix during impact. The appearance of film-like RA between the martensite laths will help improve the toughness of the matrix and delay the cracking of the interface.

However, as tq increased to 120 s, it can be seen from Figure 10c that the same dimples exist in the dendritic region, but the eutectic region is composed of a large number of smooth regions, which is different from Figure 9b. In addition, the impact toughness decreases to 6.3 J·cm^−2^. This is related to the hardness decrease of the matrix, which is caused by an excessively high fraction of RA (19.4%) in it, which leads to the partial boride that can hardly be supported by the matrix. Therefore, boride will fall off from the interface, resulting in a large number of brittle fractures in the eutectic region.

#### 3.3.2. Effect of tq on the Wear Property of High Boron Steel

Q&P heat treatment is beneficial for the improvement of the wear resistance of high boron steel. Mass loss records and measurement deviations after a dry sliding wear test are plotted in Figure 11. The deviation of mass loss of different samples and reference specimens (NM500) fluctuates between 0.001 or 0.002 g, which proves the validity of the measurement data. After sliding for 3 h, the mass loss of the as-cast sample is 0.084 g, which is lower than NM500 (0.124 g). The mass loss of samples after Q&P heat treatment are 0.06 g (tq = 30 s), 0.067 g (tq = 60 s), 0.065 g (tq = 90 s), and 0.075 g (tq = 120 s), and that of all the samples are below that of the as-cast sample.

The wear resistance of high boron steel is affected by the matrix microstructure. The sample with tq of 30 s shows better wear resistance due to the higher hardness of the matrix. However, as the tq increases, the fraction of RA in the microstructure gradually increases to make the hardness of the matrix decrease. The decrease of the matrix hardness causes a larger wear loss. Besides, the mass loss of the sample with tq of 90 s is lower than that of the sample with 60 s, which is likely due to the martensitic transformation from film-like RA caused by the wear stress in the surface layer.

The transformation during the wear process can be demonstrated by Figure 12. In the as-cast sample, the brittle spalling pits and grooves along the sliding direction are observed on the worn surface, shown in Figure 12a. It can be seen from Figure 12d that the coarse borides are broken in the worn subsurface on the section microstructure. Jian et al. [36] demonstrated that Fe_2_B can hardly be deformed due to its high hardness and elastic modulus but will causes brittle spalling from the matrix during the wear process. Few pits and grooves are observed on the worn surface of the samples with tq of 90 s, as shown in Figure 12b, and a thicker deformation zone is observed in the section microstructure (Figure 12e). Although they are broken, the borides are deformed along the sliding direction, making the clear flow lines appear in the deformation zone.

The worn surface of the sample is affected by both the compressive stress from the load and the shear stress of the lower specimen during the wear process. Berkowski et al. [37] demonstrated that carbide microcracks are the center of fracture nucleation as well as fatigue spalling in ledeburitic chromium steels. For the as-casted high boron steel, the boride is coarse and concentrated with clear microcracks, and has a long and unstable interface with the matrix, as shown in Figure 12d, which will cause brittle spalling of the martensite matrix under the shear stress from the lower specimen. The boride without support from matrix will be exposed on the worn surface, and will be gradually broken and peeled off under the compressive stress and shearing stress, resulting in a large amount of mass loss. A thicker deformation zone is observed in the worn subsurface layer of the sample with tq of 90 s (Figure 12e), in which there are clear flow lines and doped with many boride particles. RA will induce martensite nucleation under plastic deformation, and the phase transformation strengthening (TRIP effect) will increase the toughness of the matrix to better support the boride particles. In addition, many non-shedding oxidized wear adhesive particles are produced on the worn surface during the wear process (Figure 12b), which will hinder the wear process and improve wear resistance of the steel. It is comparable to the results of Kazimierz et al. [38] on the tribological behavior that AlCrSiN-Coated Tool Steel K340, that is, a large number of relatively large carbides constitute a natural obstacle to the counter body material, and the adhesive wear makes the wear rate of the sample lower than the abrasive wear, as demonstrated in the literature [39].

However, RA in an excessive amount is not beneficial to the wear resistance of the high boron steel. The peeling pits appeared on the worn surface since the sample with tq of 120 s contains 19.7 vt.% of RA, as shown in the dashed box in Figure 12c. A few deformed flow lines and spalled pits are observed within a thinner deformation zone on its section microstructure (the dashed box in Figure 12f). During the wear process, RA in the surface layer is worn off because of its low hardness before it transforms to martensite. Then, the matrix can hardly support the broken borides effectively. Therefore, it increases the mass loss of the sample.

## 4. Conclusions

The usual toughening methods for high boron steel, such as alloying and high temperature heat treatment, mostly focus on the improvement of boride morphology, but have a very limited effect on enhancing the toughness of the matrix. Inspired by the production of Q&P steel, Q&P heat treatment was used for high boron steel in this work. The results show that microstructural and mechanical properties, especially toughness and wear resistance of the steel, have been improved successfully after being treated because this process can only ensure the improvement of boride morphology by austenitization. However, the process also enables more fraction RA in the matrix that can be stabilized to room temperature through the steps of quenching and partitioning. It will provide guidance for production and the optimization of a heat treatment process of high boron steel.

The insights regarding the present study have been obtained as the following.
(1)The microstructure of as-cast high boron steel is composed of pro-eutectic austenite dendrites and eutectic, which is from the reaction of L→ γ + Fe_2_B. After solidification, the transformation will occur in the austenite. The borides in eutectic show net-like and rod-like morphology without change in cast cooling and, finally, the matrix is composited of martensite laths and RA.(2)After being treated by the Q&P process, the borides are partially broken and spheroidized successfully. The matrix is composed of UM and TM and the morphology of RA is mostly changed from blocky to film-like. As the tq in the Q&P process extend from 30 s to 120 s, the volume fraction of RA in the microstructure increases from 4.2% to 19.7% and the distribution becomes more uniform.(3)Although the macro-hardness of the samples gradually decreases from 61.8 to 55.3 HRC with the extension of tq, the toughness is significantly improved, compared with the as-cast sample, and reaches the maximum (7.5 J·cm^−2^) when tq is 90 s. In a dry sliding test, the mass loss of treated samples are about 0.06–0.075 g, which shows better wear resistance than that of the as-cast sample (0.084 g) and NM500 (0.124 g).(4)Mechanical properties improvement is attributable to the change of boride morphology and the improvement of the matrix by increasing the volume fraction of RA. However, as a soft phase in steel, excessive volume fraction of RA will inevitably lead to a decrease in the hardness of the matrix, resulting in deterioration of toughness and wear resistance. From the experimental results of the high boron steel, the volume fraction of RA should be controlled at about 16%.

## Figures and Tables

**Figure 1 materials-14-01556-f001:**
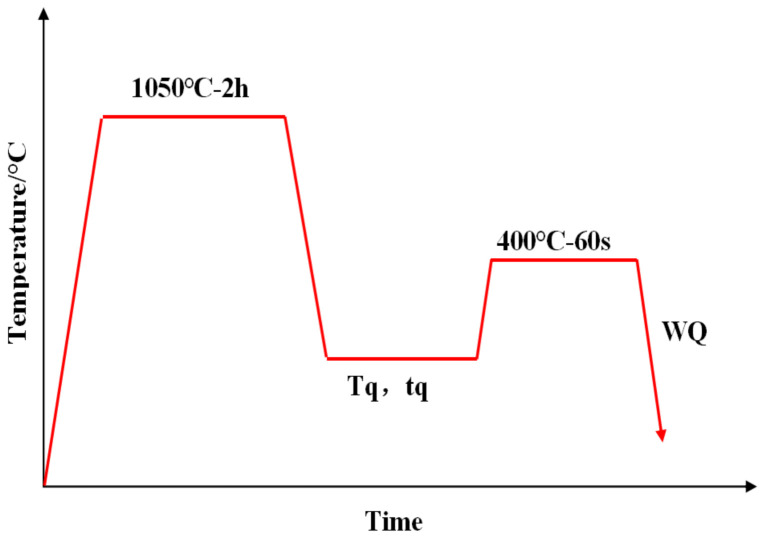
The scheme of the quenching and partitioning (Q&P) heat treatment process of the high boron steel.

**Figure 2 materials-14-01556-f002:**
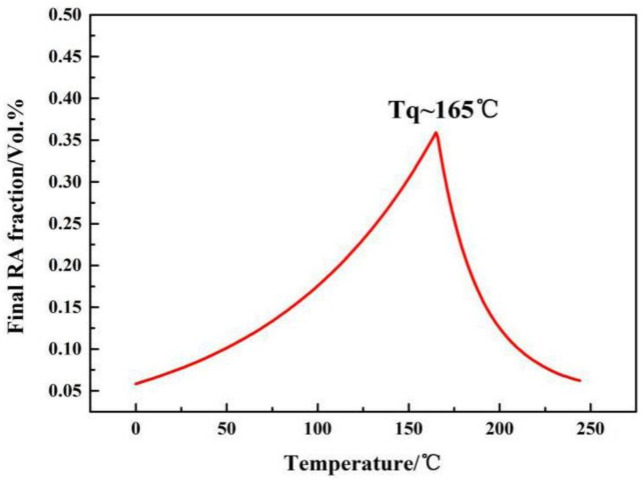
Predicted RA fraction according to Speer et al. theory.

**Figure 3 materials-14-01556-f003:**
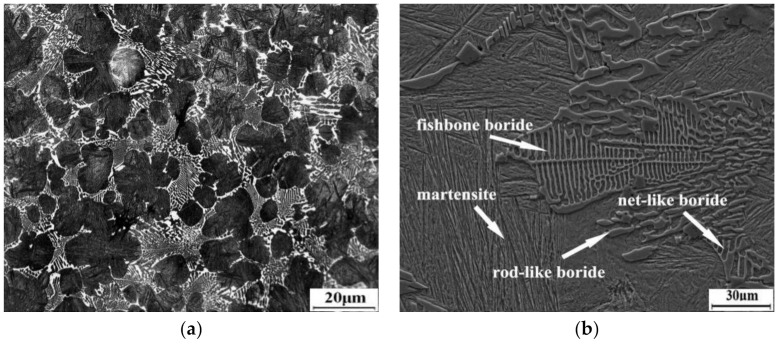
As-cast microstructure of experimental steel: (**a**) OM micrography and (**b**) SEM micrography.

**Figure 4 materials-14-01556-f004:**
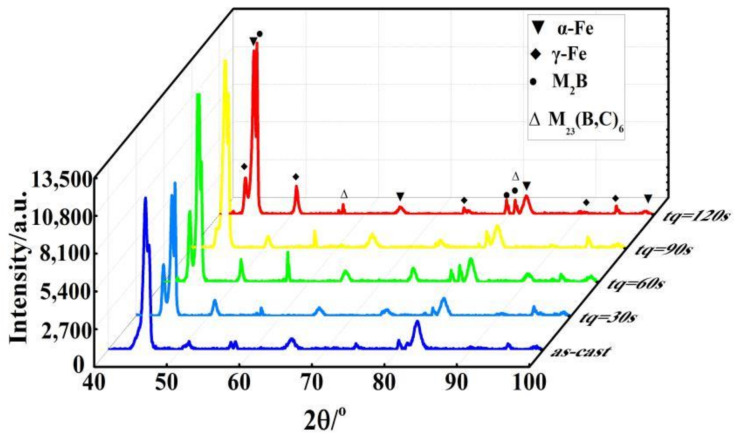
X-ray diffraction (XRD) spectra of samples in the as-cast condition and under different tq.

**Figure 5 materials-14-01556-f005:**
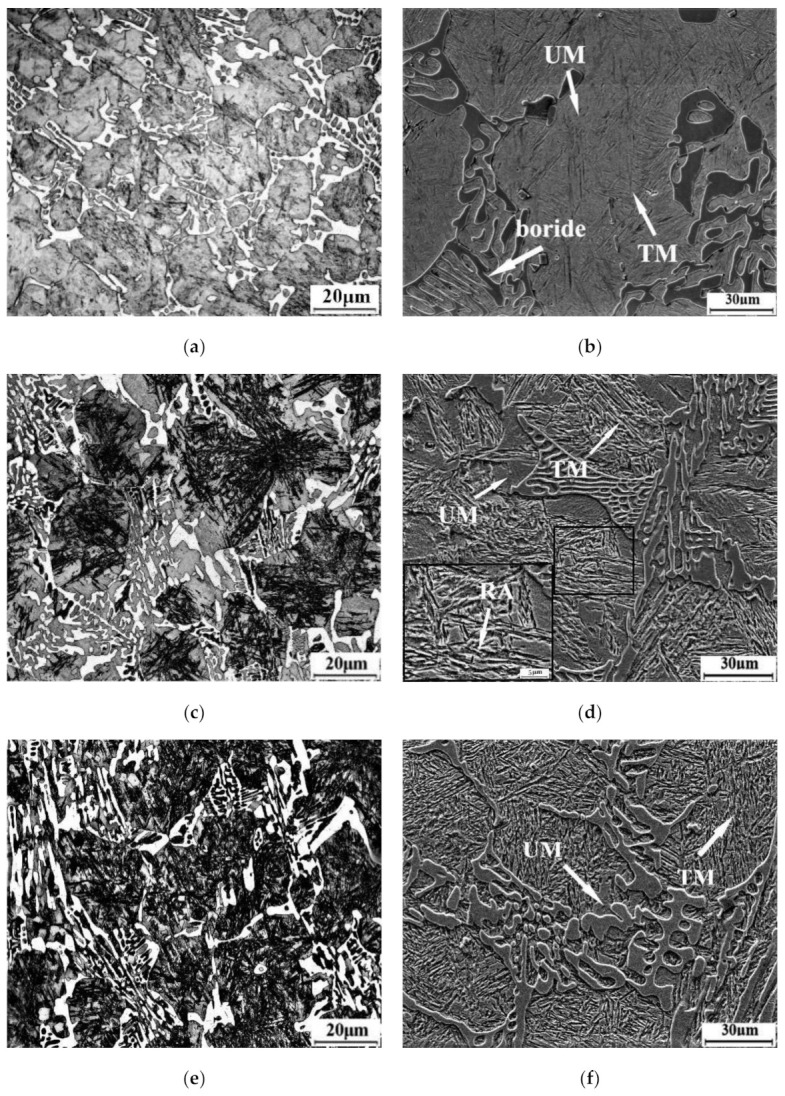
The microstructure of samples under different tq: (**a**,**b**) tq = 30 s, (**c**,**d**) tq = 60 s, (**e**,**f**) tq = 90 s, (**g**,**h**) tq = 120 s.

**Figure 6 materials-14-01556-f006:**
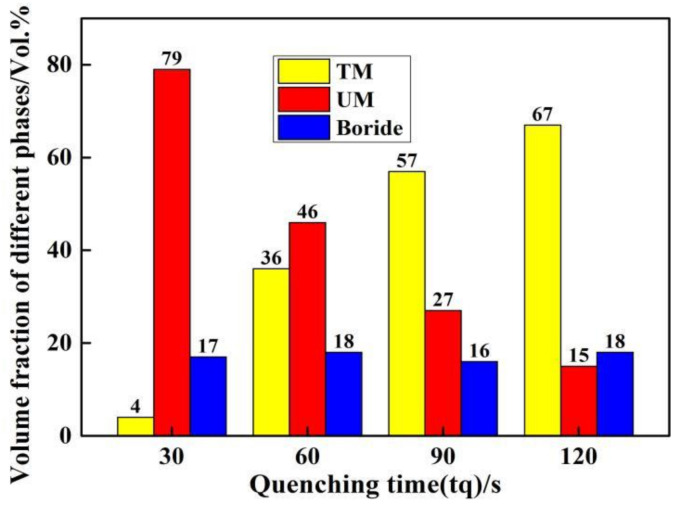
Quantitative results of samples under different tq.

**Figure 7 materials-14-01556-f007:**
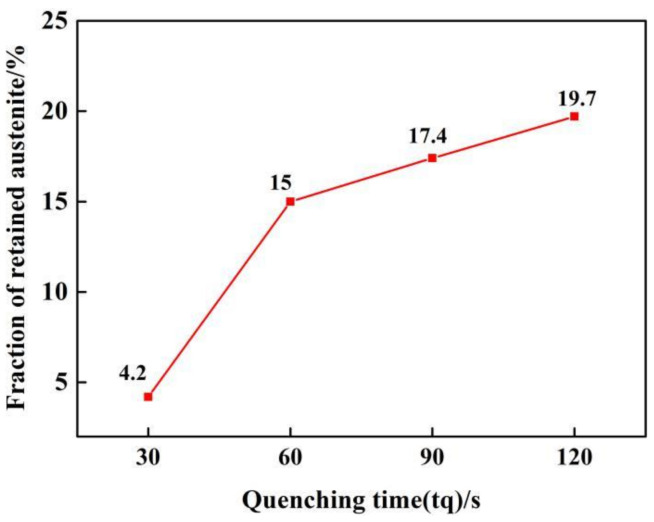
Changes in fraction of RA in the microstructure under different tq.

**Figure 8 materials-14-01556-f008:**
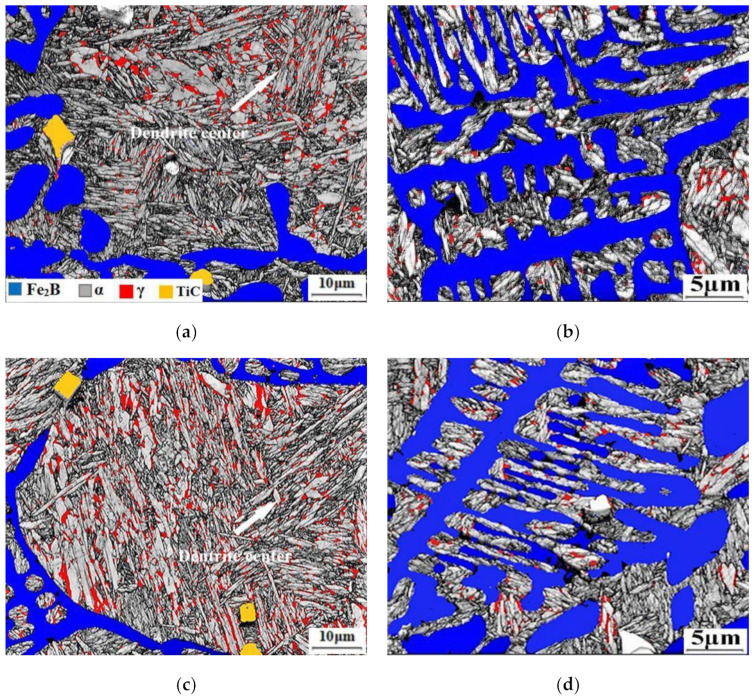
EBSD phase map analysis results of samples: (**a**,**b**) tq = 60 s, (**c**,**d**) tq = 120 s.

**Figure 9 materials-14-01556-f009:**
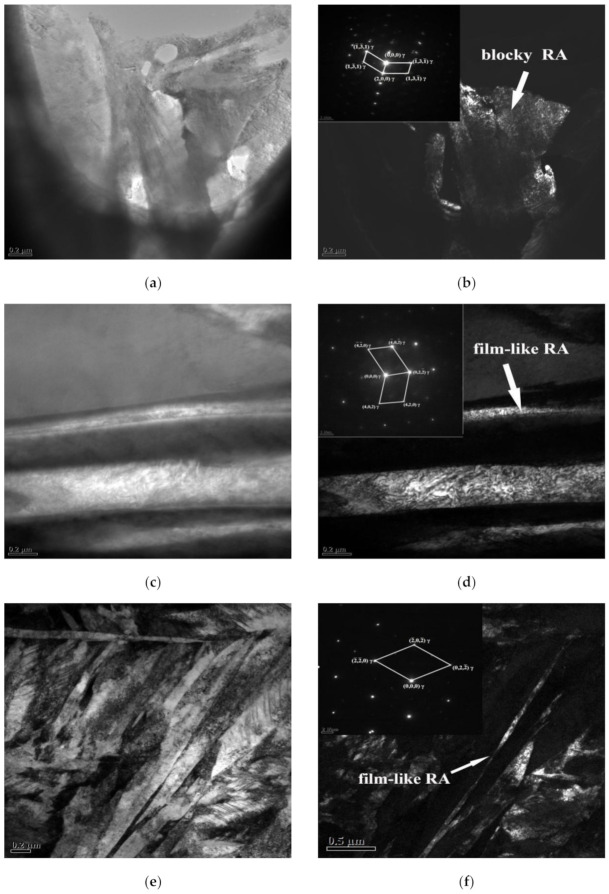
The TEM morphology of samples: (**a**,**b**) as-cast, (**c**,**d**) tq = 90 s, and (**e**,**f**) tq = 120 s.

**Figure 10 materials-14-01556-f010:**
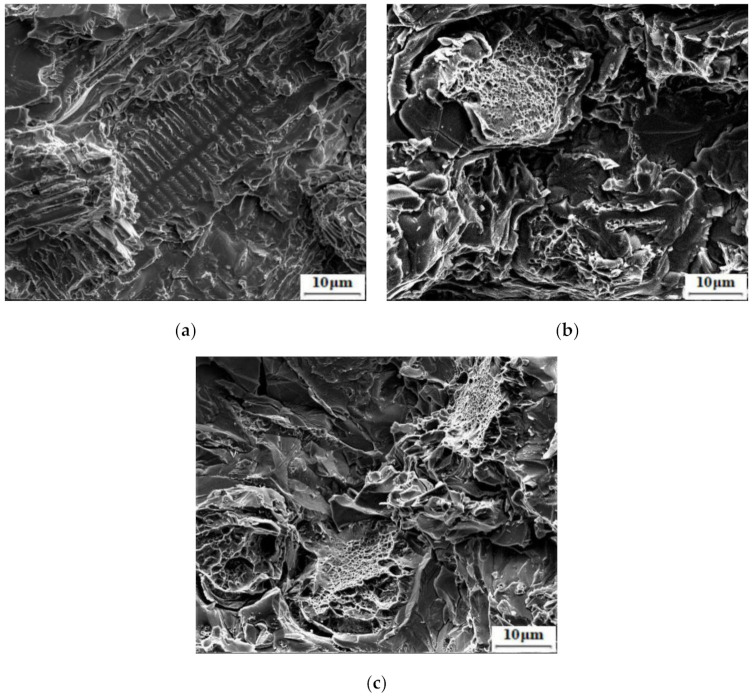
SEM fractographs of samples: (**a**) as-cast, (**b**) tq = 90 s, and (**c**) tq = 120 s.

**Figure 11 materials-14-01556-f011:**
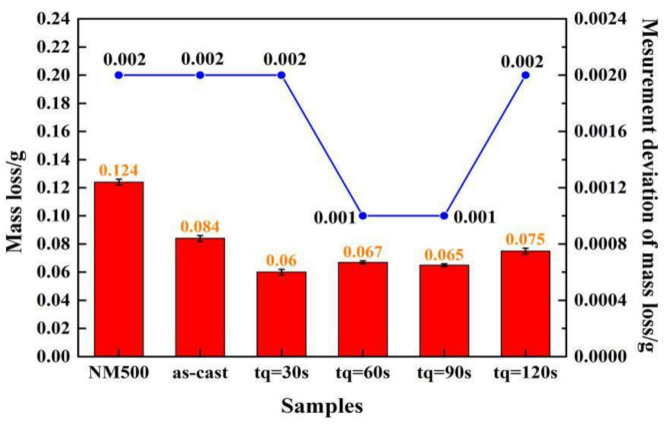
Mass loss and measurement deviation of samples and a reference specimen.

**Figure 12 materials-14-01556-f012:**
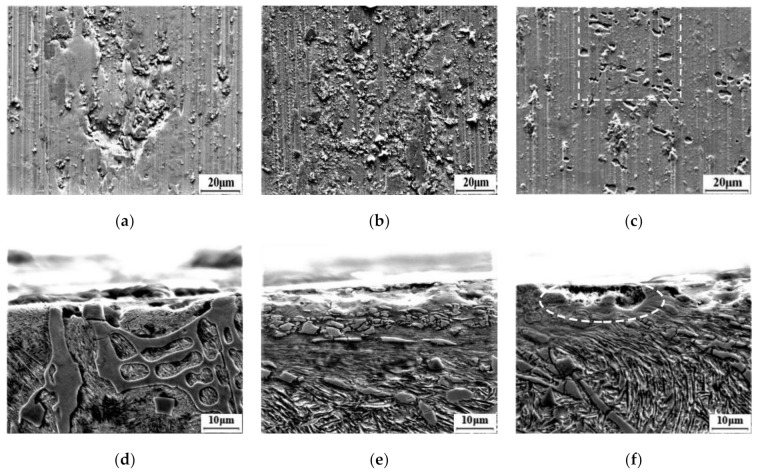
Worn surfaces and sub-surfaces morphology of samples: (**a**,**d**) as-cast, (**b**,**e**) tq = 90 s, and (**c**,**f**) tq = 120 s.

**Table 1 materials-14-01556-t001:** Chemical composition of high Boron Steel/wt.%.

C	B	Mn	Al	Si	Ti	Nb	V	S	P
0.45	1.6	4.0	0.8	1.0	0.3	0.03	0.05	<0.06	<0.06

**Table 2 materials-14-01556-t002:** The hardness and impact toughness of samples treated at different tq.

tq/s	Hardness/HRC	Impact Toughness/J·cm^−2^
30	61.8	4.4
60	58.0	5.6
90	56.1	7.5
120	55.3	6.3

## Data Availability

Not applicable.

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
