# Peer review of "The Effect of Quenching and Partitioning (Q&P) Heat Treatment on the Microstructure and Mechanical Properties of High Boron Steel"

_materials, 2021, doi:10.3390/ma14061556_

Round 1
Reviewer 1 Report
The reviewer comments of the paper «The effect of quenching and partition(Q&P) heat treatment on the structure and mechanical properties of high boron steel»
- Reviewer
The authors presented an article «The effect of quenching and partition(Q&P) heat treatment on the structure and mechanical properties of high boron steel». However, there are several points in the article that require further explanation.
Comment 1:
Introduction.
Need to improve.
Add a paragraph of the relevance of the investigated steel. What are its properties? In what products is it used? What problems exist.
After the purpose of the article, briefly describe what has been done in each section.
Comment 2:
- Materials and Methods
To improve visibility helpful redraw figures 1, 2, 5, 6, 10 in color.
The figure mentioned in the paragraph should be placed immediately after, therefore figure 1 must be placed above. Check it out everywhere in the text.
For measurement devices used in research, indicate in parentheses (manufacturer, city, country).
Comment 3:
- Results and discussion
The quality and resolution of the all figures needs to be improved.
Show table 2 correctly.
Comment 4:
Conclusions.
It is necessary to more clearly show the novelty of the article and the advantages of the proposed method. What is the difference from previous work in this area? Show practical relevance. What is the difference from other researchers?
Conclusions should reflect the purpose of the article.
The article is interesting and written at a good scientific level. Authors should carefully study the comments and make improvements to the article step by step. After major changes can an article be considered for publication in the "Materials".
Author Response
Dear Reviewer,
Thank you very much for your consideration of our manuscript. We would also like to thank you for your valuable comments and suggestions. According to the comments, we have revised our manuscript carefully and the revised manuscript is re-submitted. All the corrected parts are highlighted in the revised manuscript. The detailed response to your comments is listed in the attachment. We hope that our answers are satisfactory and that the revised manuscript can be accepted for publication. Please don’t hesitate to contact me if you have any questions.
Detailed response are submitted by attachment, please see the attachment, thank you again for your care and patience,and wish you every success in your work and a happy life.

Reviewer 2 Report
The paper describes the effect of heat treatment on the properties of boron steel. The paper could be published after improving the wear-related section. Probably the results and discussion should be improved after adding the wear-tests repetitions.
My comments on the paper:
- Q& P phrase should be explained in the abstract
- The abstract should inform about the goal of the work- please improve it.
- Please add in the paper, information about repetitions of wear tests. How many samples of each material were tested?
- Please improve the readability and quality of table 2. (some formatting issues).
- Please add in fig. 10 the results for the reference material.
- Please add in fig. 10 deviations/ min-max - now, I am worried that there is no statistical valuable difference between the tested samples.
- In the whole paper use 'mass loss" instead of " wear weight loss''.
- Authors wrote: "coarse borides with hardness of 1427HV..." - please support this phrase by literature references.
- This phrase:
" The transformation during wear process can be demonstrated by wear test with result shown
in Figure 11. In the as-cast sample, the brittle spalling pits and grooves along the sliding direction
are observed on the worn surface (Figure 11a). It can be seen from Figure 11d that the coarse
borides are broken in the worn subsurface. The coarse borides with hardness of 1427 HV can
hardly be deformed, which causes brittle spalling from the matrix during wearing. They will be
gradually broken and peeled off from the worn surface. Few pits and grooves are observed on the
worn surface of the samples with tq of 90s, shown in Figure 11b, and a thicker deformation zone is
observed in the Figure 11e. Although they are broken, the borides are deformed along the
wearing direction making the obvious deformed flow lines appear in the deformation zone. The
borides are nearly not peeled off because the transformation of RA into martensite (TRIP effect)
increases the hardness and toughness of the matrix under plastic deformation, and the matrix can
surround and support borides effectively.
"
should be removed because it is repatitions of the previously given phrase. - The paper contains many typos - please improve it in whole paper likewise: " partition(Q&P) ", " steels[3-5]"
Author Response

(The authors gave the same response as above.)

Reviewer 3 Report
Well Done, pay attention to some editing but the paper convinced me.
Author Response
Dear Reviewer,
Thank you very much for your consideration of our manuscript. We really appreciate your affirmation and valuable suggestion to our manuscript. We have carefully considered all comments and incorporated them in the revised version.

Round 2
Reviewer 1 Report
The authors have improved the article according to the comments. The article can now be published.
Author Response
Dear Reviewer,
we really appreciate your affirmation and valuable suggestion to our manuscript. We have carefully considered all comments and incorporated them in the revised version. Thank you again for your care and patience,and wish you every success in your work and a happy life!
Reviewer 2 Report
Thank you for your improvements. Your responses are acceptable and satisfy me. However, I have some minor comments regarding the wear results:
1) please do not use in the paper word "wearing" - it is not acceptable incorrect nomenclature.
These exemplary phrases: "during wearing" or "wearing direction" is not acceptable in tribology. - Authors must improve it.
2) Please add some discussion/comparison of the results with the scientific literature e.g. https://doi.org/10.3390/ma13214895 this paper presents comparable to your results where the carbides constitute a natural obstacle for the counterbody material.
3) improve the typos. missing lower case in "Fe2B" or missing spaces between the words and reference numbers eg. "austenization[23,"
Author Response
Dear Reviewer,
Thank you very much for your consideration of our manuscript. We would also like to thank you for your valuable comments and suggestions. According to the comments, we have revised our manuscript carefully and the revised manuscript is re-submitted. All the corrected parts are highlighted in the revised manuscript. The detailed response to your comments is listed in the attachment. We hope that our answers are satisfactory and that the revised manuscript can be accepted for publication. Please don’t hesitate to contact me if you have any questions.
Detailed response are submitted by attachment, please see the attachment, thank you again for your care and patience,and wish you every success in your work and a happy life!
